# Profiles of University Students Who Graduate on Time: A Cohort Study from the Chilean Context

**DOI:** 10.3390/bs13070582

**Published:** 2023-07-13

**Authors:** Ana Moraga-Pumarino, Sonia Salvo-Garrido, Karina Polanco-Levicán

**Affiliations:** 1Departamento de Administración y Economía, Universidad de La Frontera, Temuco 4780000, Chile; 2Departamento de Matemática y Estadística, Universidad de La Frontera, Temuco 4780000, Chile; sonia.salvo@ufrontera.cl; 3Programa de Doctorado en Ciencias Sociales, Universidad de La Frontera, Temuco 4780000, Chile; k.polanco01@ufromail.cl; 4Departamento de Psicología, Universidad Católica de Temuco, Temuco 4780000, Chile

**Keywords:** academic achievement, CART profiles, higher education, terminal efficiency, timely graduation

## Abstract

Graduating from higher education on time is an important topic, given its relation to students’ academic success and the efficiency of the institutions. However, a low percentage of university students finish their studies on time, which poses a challenge that requires the identification of the factors that account for this phenomenon. This study aimed to identify and characterize profiles of students who graduate on time. The population is 514 university students (45% men, 55% women), with an average age of 19.5 years (SD= 1.9) studying business at a university in Chile who belong to four cohorts entering between 2011 and 2014. The results obtained from the Classification and Regression Tree (CART) technique demonstrate eight student profiles constructed considering different variables at the pre-university, transition-motivation, and university levels. As the primary outcome, the profile of the student who graduates on time is characterized by a good performance in the first year of university, enters university right after high school, and takes advantage of institutional support by participating in academic tutoring. These findings suggest that institutions can implement specific strategies from the beginning of the university journey to promote on-time graduation.

## 1. Introduction

In recent years, universities have shown interest in studying graduation, since this is considered an indicator of the quality and terminal efficiency of the education system [1,2,3,4,5,6]. On the other hand, terminal efficiency, defined as the percentage of students who complete their program in the time allotted, is an important factor to identify the obstacles that institutions of higher learning face [3].

To improve on-time graduation, it is crucial to advance social mobility and the return on the resources invested in education by the state and families [2]. In addition, it can have a positive impact on the development of the country by increasing productivity, collecting taxes, and reducing poverty [5]. Conversely, not graduating on time can have significant emotional and financial costs, including frustration, mental health problems, the payment of additional years of tuition, and the opportunity cost of not receiving a salary according to their profession [1,2,7].

For the Organization for Economic Cooperation and Development (OECD), the graduation rate is a relevant indicator, and it requires educational interventions and public policies that improve the internal efficiency of the system. The characteristics of the students and the institution are relevant factors to understand graduation [8]. However, to reach on-time graduation, the schools need to be committed, since this depends largely on institutional factors [2,5].

In Chile, the explosive growth in higher education enrolment for more than three decades has been accompanied by low rates of permanence and graduation [1,9,10,11,12]. According to the Higher Education Information System (SIES in Spanish), used to assess the internal efficiency of higher education institutions, the first-year retention rate in programs offered by Chilean universities is 78.5% (average 2015–2019 cohorts), showing a slightly positive trend in the last 5 years [13]. However, the average real duration and over-duration in the same period rose to 12.6 years and 31.5%, respectively, values that have remained relatively stable in the last 5 years. In professional degree programs, the average real duration average is 13.2 semesters, 3.2 semesters more than the average formal duration of 10 semesters defined in the curricula [14]. On the other hand, the OECD (2019) reports that timely graduation in Chile is especially low, with only 16% of students graduating in the formal duration.

Since 2016, the public policy of free tuition as a funding system has caused concern for on-time graduation since the funding is limited to the formal duration of the program. In 2020, 55% of the total enrolment in higher education and 62% of university students had access to free tuition [1]. Therefore, in Chile the focus has shifted from coverage to the effectiveness of the training processes, focusing on desertion, permanence, and on-time graduation [10,12].

In summary, several studies have shown that a policy centered on increasing coverage and controlling desertion without a specific approach to graduation has given rise to partial advances in the terminal efficiency of universities [1,2,10,15,16]. However, few studies concentrate on the factors that influence graduation beyond the first year, and even fewer on on-time graduation [17,18,19]. Internationally, most studies have been conducted in the United States and Europe, and more research is needed on the trajectories of persistence and graduation from a longitudinal and contextualized approach.

In Chile, most studies have concentrated on over-duration and permanence/desertion, rather than on graduation in itself, and research is required that integrates sociodemographic variables and institutional support variables to identify early on those students who are most likely to graduate on time. Therefore, this study endeavors to contribute to the terminal efficiency of higher education institutions in Chile by identifying profiles of students who graduate on time, considering academic and non-academic factors related to the pre-university and university experience.

### Theoretical Framework: Terminal Efficiency and On-Time Graduation

Terminal efficiency is a concept that refers to a student’s ability to complete a certain education level satisfactorily. According to López et al. [20] and Hernández-Falcón et al. [21], this term has been studied from two main perspectives. First, from a systemic conception, terminal efficiency is considered the relation between the number of students who graduate and the resources used to obtain it, such as teaching staff and financial resources. Therefore, it is used as an indicator to compare the first-year admission rate to the graduation rate in a certain period [15,22]. Second, from systems theory, terminal efficiency has been used as a quantitative indicator to analyze institutional functioning and selectivity throughout the academic trajectory of the student population [20,23,24]. In this approach, the characteristics of the students and their education trajectories are analyzed to determine how they influence their ability to graduate on time.

When studying the terminal efficiency of institutions, it is important to consider the level of aggregation of the information available. Some studies have focused on group statistics by discipline or program [20,23]. However, Álvarez et al. [23] suggest the need to work with the individual information of each student to determine their exact situation at the time of the study. This way, the terminal efficiency of each student can be analyzed and studied individually, which becomes a dichotomous variable that takes a value of 1 if the student graduates efficiently and 0 if they do not. This obtains terminal efficiency rates by cohort, i.e., the number of students who graduate efficiently in relation to the total number of students registered in a cohort.

On the other hand, from the point of view of terminal efficiency, it must be understood that retention and graduation are two complementary perspectives that universities must understand and work on, especially for those students who have been defined as vulnerable or at risk of dropping out. Retention is related to the ability of the institution to keep students [25], whereas graduation promotes the educational trajectory concluding favorably and in a suitable time frame. It is essential to focus on how to change institutional practices to strengthen graduation, instead of changing students’ admission characteristics.

To address retention and graduation, the phenomenon should be understood from a multidimensional perspective considering the student-related factors and factors associated with the education context. This approach calls on universities to change their perspective, emphasizing their role and responsibility rather than the characteristics of the students. The interventions must focus on ensuring a process that integrates the students into the institution, especially for those universities whose students are the first generation in higher education.

Despite the limited international literature around on-time graduation, there is an interest in higher learning institutions to improve levels of excellence, recognizing the complexity of graduation processes [26]. It has been found that on-time graduation is a multifactorial process that includes individual and collective factors at the personal and institutional levels [27,28,29]. On the other hand, Vain [6] emphasizes the importance of analyzing education trajectories towards on-time graduation and considering the factors that affect students’ behavior while attending university. Related studies identify important variables for graduating, such as the parents’ socioeconomic and education levels, school origin, and gender [11,30,31,32,33,34,35,36], as well as belonging to an ethnic minority [37,38]. In addition, it has been suggested that students with weaker preparation have a lower university graduation rate and a higher dropout rate [11]. However, students’ initial preparation and other entry conditions only partially explain low graduation, as resources and institutional support strategies have a significant impact on university student success [11,31,34].

Moreover, on-time graduation has been related to other student sociodemographic characteristics such as place of residence [32,37,38,39], gender, although the research is inconclusive [33,37,40], age and marital status [40], and family characteristics associated with the support provided and education level [30,36,40,41]. In addition, academic factors prior to university entry have been identified, such as high school grades [30,32,40,42,43] and admission test results, particularly in mathematics and language [30,38,44]. Nevertheless, some studies such as Al-Nassar et al. [45] do not agree with this assertion, maintaining that academic information prior to university is not relevant to their research.

On the other hand, program and university-related factors have an impact on on-time graduation. These include in particular academic support [41,44] and financial support programs [30,38], approved credits [18,32,37,41], especially in the first year [38,41,44], and the first-semester grade average [37,38,40]. The approval of key subjects for each program is also relevant since this could mark the student’s trajectory in terms of timely graduation [45]. In addition, a student’s persistence and motivation, especially in the first year, are important [46]. Self-esteem, well-being, and commitment positively influence the student’s academic performance [47,48], whereas academic procrastination reduces it [49].

In a recent investigation with students at an Australian university [50], gender, university entrance score, number of majors in different disciplines, and second- and third-year academic performance predicted on-time graduation. Another study reports that first-year academic performance was higher for those students who graduated on time, considering that the grade average in each year was positively correlated with the grade average in the following year. Moreover, the previous education level and work experience predicted the grade average obtained at the end of the program [51]. For their part, Alyahyan and Düştegör [52], based on a full review of related studies, conclude that the main factors that influence on-time graduation are academic, pre-university, and university performance, and demographic aspects present in 69% of the reviewed works.

In Chile, research into on-time graduation is limited. Pey et al. [53] found that the high academic load and bureaucracy in graduation activities are institutional factors affecting the problem. On the other hand, Carvajal et al. [54] identified personal characteristics, initial academic performance, final academic performance, and teaching as relevant factors in over-duration in various university programs. For their part, Von Hippel and Hofflinger [10] suggest that to improve graduation and persistence in Chile it is necessary to use data and analysis to identify the students at risk of dropping out, to guide the interventions, and evaluate their effectiveness. According to these authors, the data available during enrolment are weakly predictive of persistence, and this improves significantly once data on grades in the first years are incorporated. Furthermore, they agree on the importance of analyzing and assessing based on data to make informed decisions that contribute to increasing university graduation rates.

In view of the implications of prolonging the amount of time students remain in their degree program, it may be concluded that on-time graduation is essential for students and their families, the institutions, and society as a whole. The factors that would explain on-time graduation come from different contexts, i.e., they are personal, social, economic, and educational factors. In this sense, the scientific literature shows that the pre-university factors that might be associated with on-time graduation are gender, ethnicity, parent education levels, academic performance in secondary school, and school origin. On the other hand, there are factors connected to the transition between secondary school and university, such as the place of preference of the program the students entered, and the time between graduating from secondary school and entering higher education. Furthermore, as previously mentioned, university-related factors were observed, including the relevance of academic performance and participation in tutoring, among others (Figure 1). Therefore, this study aimed to identify and characterize profiles of Chilean university students studying business from four cohorts entering between 2011 and 2014 who graduated on time.

## 2. Materials and Methods

This study is framed in a positivist paradigm, using cross-sectional quantitative tools with explanatory reach and the design of which is non-experimental [55].

### 2.1. Analysis Unit

The study sample is 514 students from four cohorts (2011 to 2014) who entered the first year of business programs in a state university in Chile through regular admission. These students were followed throughout their academic trajectory for six years (5-year program plus one), which is why the data include records from 2011 to 2016, for the first cohort, and from 2014 to 2019 for the last cohort.

The regular admission process in Chilean universities is unified and selective, according to the Office of the Deputy Minister of Higher Education. The students who fulfill the minimum requirements and complete their application are evaluated on the basis of their performance in different access instruments, such as High School Grades, Grade Ranking, and University Admission Tests. These admission tests are obligatory and consist of Language and Mathematics for all the years of study. This selection process seeks to choose the best candidates for the different programs offered in Chilean universities.

The data were obtained from institutional sources, secondary data, and individual student records from university entrance and throughout the education process. Institutional data are assumed to be reliable, and any error is small and has no significant effect on the study results. Confidentiality was preserved by not using personal data, and it was assumed that missing data are random and completely independent.

### 2.2. Identification of Variables and Operational Definition

The dependent variable is on-time graduation, which takes the value 1 when the student graduates on time, i.e., finishes their program in the years of formal duration (5 years) and takes the value 0 when graduation is not on time. The explanatory variables (see Table 1) are organized in “pre_university” (includes all the academic and sociodemographic variables prior to university entry), “transition/motivation” (includes variables identified in the high school/university transition), and “university” (includes academic and institutional integration variables in the first year).

### 2.3. Data Analysis

First, an exploratory data analysis was performed to know the education trajectories of the cohorts in the study. Then, to establish the profiles of students who graduated on time, the multivariant CART analysis was used [56]. CART analyzes associations between students’ entry conditions and their education trajectories at university with on-time graduation. As stated by Ma [57], “CART creates binary divisions of groups successively based on a statistical criterion”.

Some characteristics and useful advantages of CART [58] are that it is non-parametric and, therefore, is not based on data that belong to a particular type of distribution; it is not significantly affected by atypical values in the input variables; it minimizes the likelihood that an important structure in the data set will be missed by stopping too early; it incorporates both tests with a test data set and cross-validation to evaluate the goodness-of-fit more accurately; it can use the same variables more than once in different parts of the tree. Finally, CART can be used with other prediction methods to select the set of input variables. In this document, we refer to the terminal nodes simply as “group”. Each of these groups can fully describe the characteristics of the individuals, and each profile can have an estimated average of the result [59,60]. To better interpret the results of the CART, we defined profiles with the characteristics of the students associated with the representation group [59,61]. The CARTs were obtained using the JMP^®^ [62].

Once the profiles are identified, the analysis is complemented with inferential analysis, Student’s *t*-test, and Pearson’s chi-squared test with corrected typified residues de Haberman [63] to identify categories of significant variables that help examine the identified profiles in greater detail.

## 3. Results

Next, the results are presented. The exploratory analyses were conducted considering pre-university, transition-motivation, and university variables, followed by the description of the profiles obtained by the CART analysis.

### 3.1. Exploratory Analysis

An analysis of the education trajectories of the students (Table 2) reveals that only 28% graduated on time (in 5 years), while the remaining 72% presented different situations. Of that percentage, 20% graduated with a one-year delay (t + 1), 39% dropped out of the program, and 17% are still enrolled in the university without having graduated after 6 years since admission. In addition, when analyzing the different cohorts, there was a fluctuation in the on-time graduation, ranging between a minimum of 20% (2011 cohort) and a maximum of 33% (2012 and 2014 cohorts). Variations are also noted in graduation with a one-year delay and the permanence of students in the university after 6 years since admission. On the other hand, the desertion rate remained relatively stable throughout the period studied, reaching 40% in the three last years.

Table 3 presents a synthesis of the distribution of variables related to graduation in the studied cohorts. Understanding these variables can help identify factors that affect the on-time graduation of university students.

In terms of the pre-university variables, most students come from subsidized private schools or public/municipal schools, are women, and a low percentage of their parents have a university education. In addition, almost 28% of the students come from professional-technical schools.

In terms of the transition and motivation variables, most students entered the program as their first option and began their university studies immediately after high school.

Regarding the university variables associated with academic performance, the students had an average performance higher than 4.0 in the first two semesters and an average performance below 4.0 in mathematics. In addition, almost 25% of the students participated in academic tutoring during their first year and most reside with at least one of their parents during their university studies.

Statistically significant differences were found (*p* < 0.05) in 12 of the 18 analyzed variables between those who graduate on time and those who do not. Among the pre_university variables, those that graduate on time have better high school grades (HS_GPA) and a higher SAT_Math score, are not of Mapuche ancestry (D_ETH), and are mainly women. Regarding the transition/motivation variable, those who graduated on time entered the program one-year maximum after finishing high school. With respect to the university variables, those who graduate on time achieve better academic results, participate in academic tutoring, do not work, and reside with both parents (D_LWP).

### 3.2. Profiles

The CART analysis revealed that six variables were significant in explaining on-time graduation, with 67% of them referring to the university. Of the 18 variables available in the database, University_GPA1 and University_GPA2 stand out as the most important, contributing jointly with 79.5% of the explained variance of the dependent variable. The results obtained are noteworthy since the explained percentage of the variance of the model (33.9%) was considered high compared to previous studies in this area [64,65].

Table 4 shows the variables in order of importance, indicating the explanation percentages that represent the relative relevance of each factor in the model.

Figure 2 shows the Classification Tree of on-time graduation.

The CART analysis identified eight groups of students differentiated by their ability to graduate on time. The root node represents the total population (514 students), of which only 28% graduated on time. The root node was divided into two child nodes according to the value of the University_GPA1 variable. The left child node, with a high University_GPA1 (equal to or greater than 5.0), groups 37% of the population and is associated with high performance and high on-time graduation. This node gave rise to terminal groups 1, 2, 3, and 4. The right child node, with a low University_GPA1 (below 5.0), is associated with a middle or low performance and low on-time graduation, and groups 63% of the population. This node gave rise to terminal groups 5, 6, 7, and 8. Generally, groups 1 and 2 have a higher-than-average on-time graduation, whereas groups 4, 6, 7, and 8 have a lower-than-average on-time graduation, with group 7 standing out for having no student graduating on time. Table 5 summarizes the characteristics of each of the terminal groups.

When analyzing the variables identified by CART with greater presence in the high on-time graduation profiles (>50%) graphically, the gap in graduation between the groups and cohorts in this study over time is noted.

Figure 3 shows that when the students obtained a University_GPA1 equal to or greater than 5.0, the rate of on-time graduation is more than 35% higher (2011: 34.6%; 2012: 68.8%). Figure 4, on the other hand, shows that when the students obtained a University_GPA2 equal to or greater than 4.5, the on-time graduation rate is more than 25% higher (2011: 25.8%; 2014: 44.9%). These differences in on-time graduation indicate that students with better performance in the first year at university are more likely to graduate on time.

Figure 5 shows the difference in the on-time graduation rate between students who made the transition from high school to university in less than three years, and those who took three years or more to make this transition. The gap between the two groups has increased over time, from 4% in 2011 to 27% in 2014. These results suggest that the time students take to enter the program has an important effect on their ability to graduate on time.

To complement the previous analysis, and thus identify possible courses of action to strengthen on-time graduation, we wonder what variables affect students achieving a University_GPA1 equal to or greater than 5.0. The statistically significant variables (*p* < 0.05) that contribute to a University_GPA1 greater than or equal to 5 are a higher SAT_Math, higher HS_GPA, residing with their father and/or mother during their time at university, and being a woman.

## 4. Discussion

On-time graduation in higher education is an increasing challenge due to its social and economic importance for students, their families, the country, and society in general. Several studies [1,2,3,4,5,6,7,8,9] indicate that the graduation rate is low in higher education, especially in Latin American countries such as Chile, where fewer than 20% of students graduate on time [8].

To analyze the profiles of students who graduate on time in greater detail, it is important to perform a multidimensional analysis that integrates academic and non-academic variables [2,5,20,38]. This study will enable the development of contextualized interventions that enhance educational procedures in line with the identified profiles [61].

According to the results obtained through the CART analysis, six variables are significant in identifying profiles of students who graduate on time. These variables are: academic performance during the first year at university (University_GPA1, University_GPA2, and Math_GPA1), the time between the high school/university, and participation in academic tutoring. These findings are consistent with what has been reported in the literature, which demonstrates that on-time graduation is a process that responds to multiple factors, both personal and institutional [2,5,11,31,34,35,66]. It is important to consider these factors to design specific interventions that improve terminal efficiency in higher education.

The literature and the results are in agreement that the performance in the first year of university is crucial for on-time graduation, with University_GPA1 ≥ 5.0 being the most important variable, and then University_GPA2 ≥ 4.6. This finding confirms international studies that recognize that the first year of university represents a critical school-university transition, and that lays the foundation for later academic success, persistence, and graduation [10,38,52,54]. In addition, the importance of STEM stands out, such as mathematics in the first semester, generating a positive and containing effect of on-time graduation in case the performance in the first semester is low [17].

After conducting complementary analyses to understand what variables explain a good performance in the first university semester, specifically average grades greater than or equal to 5.0, it was found that the differentiating characteristics among those who perform better are a higher score on the SAT_Math, a higher HS_GPA, being a woman, and living with their father and/or mother during their time at university. These findings support the importance of the academic performance of students in high school, as has been indicated in other studies [67]. In addition, this corroborates that high school grades have a significant impact on timely graduation since they help explain academic performance in the first year [52]. Moreover, this supports that the inequalities and differences in education opportunities and standards at the secondary level extend to higher education [68].

Regarding gender and residing with parents during university life, our findings are consistent with the existing literature on gender inequalities and the importance of family containment in higher education. In particular, it was found that being a woman is related to better performance during the first semester at university [17,38,69,70], as is living with parents, which underscores the importance of family containment during the university stage, especially in emotional and financial matters [25,69].

In addition to a good performance in the first and second semesters, another relevant variable to graduating on time is the time of transition between high school and university. Specifically, the findings make it possible to state that students who entered university immediately or a maximum of one year after finishing high school were more likely to graduate on time than those who entered later. This finding is consistent with previous studies that indicate that students who begin university at a more advanced age may be more vulnerable to not graduating due to opportunity costs and other responsibilities [17,70].

Additionally, participation in academic tutoring support is also a relevant factor in on-time graduation. Although it is secondary to the performance in the first year and high school/university transition, participating in this tutoring during the first year of university boosts on-time graduation among students performing well in the first and second semesters of the program who entered university early. This conclusion agrees with the suggestion made in several previous studies that highlight the efficacy and importance of academic support mechanisms in graduating on time [17,44]. In addition, it supports the assertion by Tinto [71] with respect to the importance of active participation in academic support services to achieve better results, boost integration, involvement, and recognition of the commitment of the institution to the education process. In addition, if this tutoring is imparted by peers, i.e., students in higher classes, they can also generate contact networks, live university life more thoroughly, and understand their place on campus [11]. Thus, the results corroborate the importance of academic tutoring for integration into university life and student success [44,71] and, therefore, it must be recognized as a key strategy that not only improves retention and increases performance, but is also fundamental to graduation, especially for economically vulnerable and first-generation students [39,72].

Finally, it was found that self-declaration of Mapuche ancestry (D_ETH) combined with low performance in the first year may have a negative impact on a timely graduation. This finding, on the one hand, is in line with several studies [37,38]; however, it requires further investigation. The differences found may be connected instead with the differences in academic entry conditions or other aspects that affect the university transition as a result of indigenous peoples being rendered invisible in the academic context.

Regarding the implications of this study, it is worth noting that the findings demonstrate the relevant factors for business students to graduate on time, contributing to decision-making at the levels of universities, public policies, and society. The results show that the most relevant factors for graduating on time are university-related; therefore, universities play an important role, offering the support necessary for students to achieve adequate performance in the first year, collaborating so that young people adapt appropriately to the new stage in their life and the context in which they will develop for several years. In this sense, since students face challenges and must perform in both theoretical and practical instances, it would be important for them to have multidisciplinary teams focused on various social, emotional, and academic areas, in constant coordination with the faculty. This way, students and their families are not burdened with additional costs while universities improve their terminal efficiency.

The limitations of this study include the sample including solely business students at a specific university; therefore, the results cannot be generalized to other populations or contexts. Moreover, the absence of financial variables limits the interpretation of certain findings. For future studies, it is recommended to consider broadening the disciplinary areas, as well as complementing them with qualitative data to delve more deeply into whether the expectations regarding graduation, study habits, and other variables of the university experience are relevant for students, and thus gain a greater understanding of the factors that explain on-time graduation in higher education institutions.

## 5. Conclusions

The results of this study made it possible to identify and characterize students who graduate on time. This is considered fundamental, keeping in mind the increase in higher education enrollment, the low on-time graduation rates, and the socio-economic difficulties in Latin America. On-time graduation favors students’ social mobility, and reduces poverty, lessening the socio-emotional and financial impact of extending the years students spend in school. The conclusions of the study emphasize the importance of implementing strategies and approaches adapted to the needs and characteristics of the institutions and their students to improve terminal efficiency.

It was identified that the average performance in the first year and the performance specifically in mathematics were the main factors in explaining on-time graduation, in addition to participation in academic tutoring supporting the performance students can achieve in their university studies. According to this, it is important for universities to consider investing in and reinforcing their early warning systems to identify potential students at risk of not graduating on time, observing entry conditions, and monitoring the performance in the first year in a timely fashion, starting from the first semester in both general and key subjects of the curriculum such as mathematics. At the same time, institutions must implement and/or bolster academic support strategies in the first year; not only must they be focused on a wider offering of mentoring and services, but they must also be aware and guarantee that most students know about these programs and their potential for graduating on time. Therefore, the terminal efficiency of institutions of higher learning involves a combination of strategies and approaches adapted to the needs and characteristics of the institution and their students, where a relevant indicator is on-time graduation.

Finally, among the contributions of this study, the usefulness of the CART method to help in identifying early on the profile of students who graduate on time stands out, where it has become clear that performance in the first year is key, especially the performance in the first semester.

## Figures and Tables

**Figure 1 behavsci-13-00582-f001:**
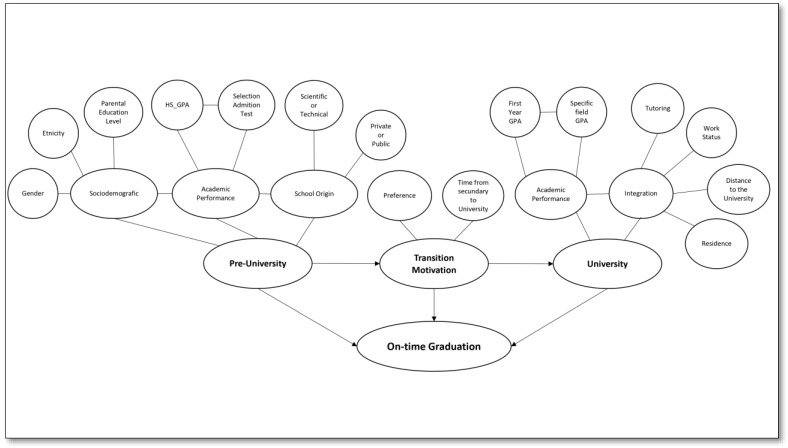
Conceptual framework.

**Figure 2 behavsci-13-00582-f002:**
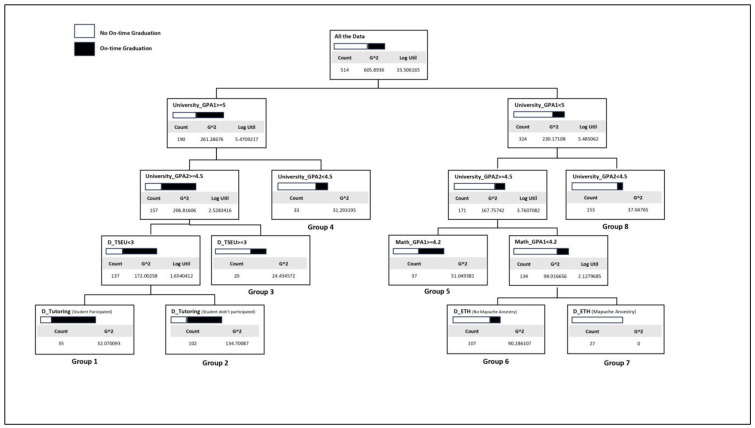
Classification Tree (CART) of on-time graduation performance, 2011 to 2014 cohorts.

**Figure 3 behavsci-13-00582-f003:**
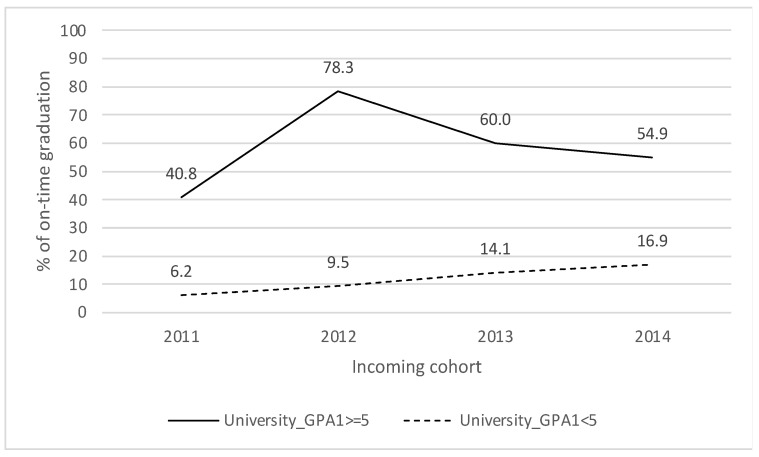
Weighted grade average First Semester and on-time graduation.

**Figure 4 behavsci-13-00582-f004:**
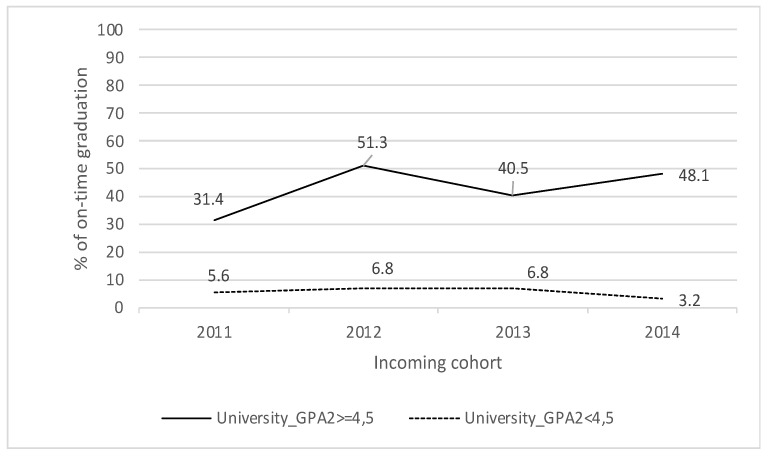
Weighted grade average Second Semester and on-time graduation.

**Figure 5 behavsci-13-00582-f005:**
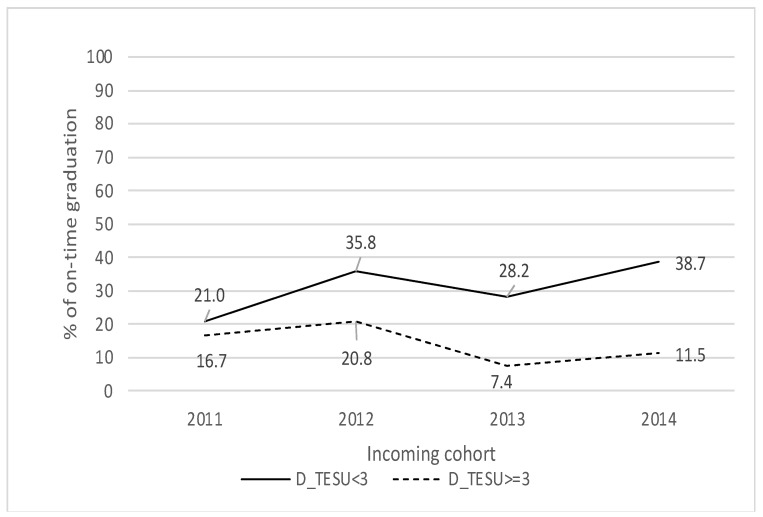
Entry to the Program and on-time graduation.

**Table 1 behavsci-13-00582-t001:** The variables included to determine on-time graduation profiles. Source: Prepared by the authors.

Pre-University Variables	
HS_GPA	High School Grade Ponderated Average (GPA)
SAT_Math	University Selection Mathematics Test Score
SAT-Lang	University Selection Language Test Score
D_STSchool	Establishment of origin: Scientific or Technical School, where: D_SS = 1, if it is scientific school; 0, other; D_TS = 1, if it is technical/vocational school; 0, other
D_FTSchool	School Type, where D_PrivateS = 1, if it is private school; 0, other; D_PublicS = 1, if it is public school; 0, other; D_SubS = 1, if it is sub-sidized school; 0, other
D_ETH	Self-declared Etnicity, where: 1 = Mapuche Ancestry; 0 = No Mapuche Ancestry
D_Gender	Gender of the student, where: 1 = Female; 0 = Male
D_EduPLevel	Education level of the parents, where: 1 = Father and/or mother with university education; 0 = Father and mother without university education
** Transition/Motivation Variables **	
D_PEC	Preference with which a student enters a career, where: 1 = first preference; 0 = not first preference
D_TSEU	Time from completion of secondary education and entry into the university, where 1, if enters into the program the following year after leaving high school or at the latest the following year (<3); 0, entry after two years (≥3)
** University Variables **	
University_GPA1	First semester university GPA
University_GPA2	Second semester university GPA
Math_GPA1	GPA in mathematics for the first semester of the degree
Math_GPA2	GPA in mathematics for the second semester of the degree
D_Tutoring	Participation in academic tutoring, where: 1 = participated; 0 = did not participate
D_EWS	Work status during studies, where: 1 = does not work; 0 = works
Distance	Distance between the university commune and origin (kilometers)
D_RDS	Residence during their studies, where: D_LWP = 1, if resides with both parents; 0, other; D_LWFoM = 1, if resides with mother or fa-ther; 0, other; D_LWR = 1, if resides with relatives or spouse; 0, other; D_LWI = 1, if resides in a room or independent.; 0, other

**Table 2 behavsci-13-00582-t002:** Education Trajectory of Students, 2011 to 2014 Cohorts. Source: Prepared by the authors.

Education Trajectories	2011–2014	2011	2012	2013	2014
N° students enrolled	514	118	133	144	119
Graduated on time (5 years)	28%	20%	33%	24%	33%
Graduate with 1 year delay (6 years)	20%	24%	17%	21%	18%
Stay in university without graduating at 6 years from entry	13%	20%	10%	15%	9%
Desert	39%	36%	40%	40%	40%

**Table 3 behavsci-13-00582-t003:** Distribution of statistically significant variables and differences between those who graduate on time and who do not graduate on time, cohort 2011 to 2014. Source: Prepared by the authors.

Variables	Range	Total	Graduate on Time	Do Not Graduate on Time
		% or Mean (SD)		
**Pre-University Variables**				
HS_GPA (*)	352–799	594 (81)	633 (77)	580 (78)
SAT_Math (*)	448–741	578 (45)	591 (48)	574 (43)
SAT-Lang	384–797	555 (61)	556 (66)	556 (59)
D_SS	0–1	72%	75%	71%
D_TS	0–1	28%	25%	29%
D_PrivateS	0–1	3%	3%	3%
D_PublicS	0–1	33%	35%	32%
D_SubS	0–1	65%	63%	65%
D_ETH (*)	0–1	18%	13%	21%
D_Gender (*)	0–1	55%	63%	52%
D_EduLevel	0–1	15%	15%	16%
** Transition/Motivation Variables **				
D_PEC	0–1	82%	88%	80%
D_TSEU (*)	0–1	82%	91%	78%
** University Variables **				
University_GPA1 (*)	1.0–6.6	4.67 (0.80)	5.22 (0.50)	4.44 (0.79)
University_GPA2 (*)	1.0–6.5	4.56 (0.86)	5.07 (0.52)	4.33 (0.89)
Math_GPA1 (*)	1.0–6.8	3.99 (1.19)	4.80 (0.93)	3.65 (1.12)
Math_GPA2 (*)	1.0–7.0	3.84 (1.13)	4.45 (0.97)	3.55 (1.08)
D_Tutoring (*)	0–1	23%	31%	20%
D_EWS (*)	0–1	82%	89%	79%
Distance	0–2327	88 (204)	107 (279)	82 (166)
D_LWP (*)	0–1	32%	40%	28%
D_LWFoM	0–1	30%	27%	31%
D_LWR	0–1	11%	10%	11%
D_LWI	0–1	28%	24%	29%

Note: SD: standard deviation; * *p* < 0.05 statistically significant differences between those who graduate on time and who do not.

**Table 4 behavsci-13-00582-t004:** Relevant variables according to the CART analysis to explain on-time graduation, 2011 to 2014 cohorts. Source: Prepared by the authors.

Variables	% Contribution
University_GPA1	55.8
University_GPA2	23.7
Math_GPA1	9.12
D_TSEU	5.1
D_ETH	3.8
D_Tutoring	2.6

**Table 5 behavsci-13-00582-t005:** Profiles of the terminal groups on on-time graduation according to the Classification Tree (CART), 2011 to 2014 cohorts. Source: Prepared by the authors.

Group	N° Students per Group (% Compared to Total)	% That Graduates on Time (Compared to Their Group)	Characteristics of the Group
I	35 students (6.8%)	>75%	Good performance in the first semester (University_GPA1 ≥ 5.0), good performance in the second semester (University_GPA2 ≥ 4.6), is motivated to enter the program (enters immediately or maximum in the second year after finishing high school), and takes advantage of the institutional support by attending academic tutoring.
II	102 students (19.8%)	>50% and <75%	Good performance in the first semester (≥5.0), good performance in the second semester (≥4.6), motivated to enter the program (enters immediately or a maximum of one year after finishing high school), and does not take advantage of the institutional support.
III	20 students (3.9%)	>25% and <50%	Good performance first semester (≥5.0), good performance second semester (≥4.6), and with less motivation to enter the program (enters after three or more years since finishing high school)
IV	33 students (6.4%)	<25%	Good performance in the first semester (≥5.0) and low performance in the second semester (<4.6).
V	37 students (7.2%)	>25% and < 50%:	Good performance first semester (<5.0), good performance in the second semester (≥4.5), and good performance in math in the first semester (≥4.2).
VI	107 students (20.8%)	<25%	Low performance in the first semester (<5.0), good performance in the second semester (≥4.5), low performance in math in the first semester (<4.2), and not Mapuche.
VII	27 students (5.3%)	0%	Low performance in the first semester (<5.0), good performance in the second semester (≥4.5), low performance in math first semester (<4.2), and declares Mapuche ancestry.
VIII	153 students (29.8%)	<10%	Low performance in the first semester (>5.0) and low performance in the second semester (<4.5).

Note: University_GPA1: First semester university GPA; University_GPA2: Second semester university GPA.

## Data Availability

The data that support the findings of this study are not available because it is institutional data.

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
