# Peer review of "Profiles of University Students Who Graduate on Time: A Cohort Study from the Chilean Context"

_behavsci, 2023, doi:10.3390/bs13070582_

Round 1
Reviewer 1 Report
Title: the title does not include information about the quality of your sample. Given that your sample is large and belongs to 4 cohorts, it fact should be informed in the title.
Abstract: The abstract should be expanded to include the most relevant findings from the study. As the abstract is often the only part of the paper that many readers will read, providing more information would be beneficial.
Keywords: It is recommended to list the keywords alphabetically. Avoid using keywords that are already included in the title of the manuscript. By using different keywords, you increase the chances of your paper being retrieved and cited by other researchers.
Introduction: The literature review contains references that are outdated, including papers from 1999 or earlier. Incorporating relevant papers recently published in other journals on the topic is recommended. Consider referring to journals emphasizing a significant percentage of references published within the last five years.
Additionally, the literature review is too concise. The authors should include references to previous studies, particularly those recently published, that have explored relevant findings related to the tested relationships.
Furthermore, it is suggested to provide a summary of the hypotheses after the introduction, ideally accompanied by a figure or diagram for clarity.
Method: firstly, additional information should be included regarding the procedures used for data collection. Provide details on how the data was gathered, including any specific methods or tools employed. Furthermore, consider mentioning the students who were invited but declined to participate, or provide relevant information about the response rate achieved in the study. These details will enhance the transparency and completeness of the method section.
Secondly, it is unnecessary to describe each item. However, if you believe it is essential, please ensure that the variable names are translated from Spanish to English. This will facilitate understanding and consistency for English-speaking readers.
Discussion:
Firstly, it is important to align your conclusions more closely with the findings of the study. In other words, provide a clearer justification for the connection between your conclusions and the actual findings.
Additionally, consider including a section that discusses the limitations of the study, potential future lines of investigation, and the principal contributions of the research. Your paper holds significant implications for educators, psychologists, society, and policymakers. However, it would be beneficial to elaborate further on these implications to provide a more comprehensive understanding of the topic.
Minor points
Some tables need a footnote that clarifies the abbreviations (e. g. Table 5).
Some figures should be translated from Spanish for English-speaking readers.
Given the fact that Behavioral Sciences is a Psychological journal, you could consider in the discussion section other psychological articles related to a similar topic. For instance, considering the influence of academic procrastination, and self-esteem among others, you should cite: González-Brignardello, M. P., & Sánchez-Elvira Paniagua, Á. (2023). Dimensional Structure of MAPS-15: Validation of the Multidimensional Academic Procrastination Scale. International Journal of Environmental Research and Public Health, 20(4), 3201. Acosta-Gonzaga, E. (2023). The Effects of Self-Esteem and Academic Engagement on University Students’ Performance. Behavioral Sciences, 13(4), 348.Wijaya, T. T., Yu, B., Xu, F., Yuan, Z., & Mailizar, M. (2023). Analysis of factors affecting academic performance of mathematics education doctoral students: A structural equation modeling approach. International Journal of Environmental Research and Public Health, 20(5), 4518
Author Response
Dear revisor,
Thanks for your comments, which have allowed us to improve the manuscript. Different improvements have been made to the article that are in yellow.

Reviewer 2 Report
The article in general is adjusted and the topic that is addressed is new, for that reason I congratulate you, also the CART system is interesting to me, some aspects of improvements or questions that arise:
- The range of years in which the sample is collected with the current one is more than five years, for this reason I consider that the results with a more current population, perhaps would be modified or not, depending on the reality of the population , but this of the years is a remarkable issue.
- Another question is to know which statistical program is used for data analysis if it is the SPSS program and because the variables are treated as non-parametric, some type of prior statistical test was carried out that will indicate the non-parametric tests as more adjusted, because , perhaps at some point in the study the parametrics could be considered. It would be appropriate to introduce some more specific explanation of why they are not parametric and the reason that leads to this decision discarding the parametric ones.
- It would be appropriate to specify the data collection instruments and their procedure, how this research process was carried out.
- Because reasons all the results are presented in percentages.
- It would be advisable after a point to always introduce a few lines, this can be seen between points 3 and 3.1, you go from one section to another directly, it is recommended to introduce a few lines.
At a general level, I consider that the work provides relevant information, but the most important aspect is the years of data collection, which are from 12-9 years ago.
Author Response

(The authors gave the same response as above.)

Round 2
Reviewer 2 Report
The work has improved considerably, being more clarified the doubts or improvements presented, as well as the parts introduced in the theoretical framework, discussion and conclusion.
The last recommendation that I would make prior to its publication is that it would have been appropriate to introduce some lines between Table 4 and Figure 2, as well as between Figures 3 and 4, instead of explaining all the information prior to these data it would be to introduce and interpret them. once the figure or table is presented, in this way the visual information would not be so continuous without any text.